# Intermittent subthalamic nucleus deep brain stimulation induces risk-aversive behavior in human subjects

Shaun R Patel[1,2]\*, Todd M Herrington[2], Sameer A Sheth[3], Matthew Mian[1], Sarah K Bick[1], Jimmy C Yang[1], Alice W Flaherty[2,4], Michael J Frank[5], Alik S Widge[4], Darin Dougherty[4], Emad N Eskandar[1]

[1]Department Neurosurgery, Massachusetts General Hospital, Harvard Medical School, Boston, United States; [2]Department of Neurology, Massachusetts General Hospital, Harvard Medical School, Boston, United States; [3]Department of Neurosurgery, Baylor College of Medicine, Houston, United States; [4]Department of Psychiatry, Massachusetts General Hospital, Harvard Medical School, Boston, United States; [5]Department of Cognitive, Linguistic and Psychological Sciences, Brown University, Providence, United States

**Abstract** The subthalamic nucleus (STN) is a small almond-shaped subcortical structure classically known for its role in motor inhibition through the indirect pathway within the basal ganglia. Little is known about the role of the STN in mediating cognitive functions in humans. Here, we explore the role of the STN in human subjects making decisions under conditions of uncertainty using single-neuron recordings and intermittent deep brain stimulation (DBS) during a financial decision-making task. Intraoperative single-neuronal data from the STN reveals that on high-uncertainty trials, spiking activity encodes the upcoming decision within a brief (500 ms) temporal window during the choice period, prior to the manifestation of the choice. Application of intermittent DBS selectively prior to the choice period alters decisions and biases subject behavior towards conservative wagers.
DOI: https://doi.org/10.7554/eLife.36460.001

\*For correspondence:
shaun.patel@me.com

## Introduction

Deep brain stimulation (DBS) is a remarkable therapy that has revolutionized the potential for treating neurological and neuropsychiatric illness by directly modifying neural function, though the underlying mechanism of action remains unknown. In its simplest form, DBS can be thought of as a pacemaker for the brain. Current DBS devices deliver continuous electrical stimulation to a targeted brain region to modify or reset abberant neural activity or synchrony (*Herrington et al., 2016a*). A major area of current research is in identifying more refined methods of stimulation delivery by exploring the timing of stimulation delivery, multi-site stimulation, and real-time sensing and stimulation.

The subthalamic nucleus (STN) is a small almond-shaped nucleus in the basal ganglia classically known for its role in inhibiting motor responses as part of the indirect pathway (*Schmidt and Berke, 2017*; *Schmidt et al., 2013*). More recently, a growing body of literature has begun to uncover a more nuanced role of the STN in higher-order cognitive processes such as, emotional processing (*Le Jeune et al., 2009*; *Le Jeune et al., 2008*; *Drapier et al., 2006*; *Eitan et al., 2013*), response inhibition (*Frank et al., 2007*; *Cavanagh et al., 2011*; *Isoda and Hikosaka, 2008*), and even psychiatric illness (*Mallet et al., 2008*).

**eLife digest** Deep brain stimulation, or DBS for short, is used to treat movement disorders like Parkinson's disease in patients who are responding inadequately to medications. It requires implanting an electrode into the brain and using electrical stimulation aimed at a specific cluster of brain cells to reduce unwanted symptoms. DBS helps to normalize abnormal brain activity similar to a pacemaker resetting an abnormal heart rhythm. Scientists are currently studying whether DBS might also help people with obsessive-compulsive disorder, depression, Alzheimer's disease or other disorders that affect thinking.

To alter human behavior and treat disorders that affect thinking, DBS will have to be delivered at precise time points as the brain processes information. One potential target is for DBS in both movement and thinking disorders is the subthalamic nucleus. This is a small almond-shaped cluster of brain cells that helps people stop movements. Recent studies suggest it also may play a role in processing emotions, controlling inappropriate responses, and psychiatric illnesses.

Now, Patel et al. show that the subthalamic nucleus helps people decide what to do in the face of uncertainty and that targeting this brain structure with DBS can shift a person's decision-making. In the experiments, patients with Parkinson's disease who were awake and undergoing surgery to implant the DBS electrodes also played a computerized gambling game. Patel et al. recorded the electrical activity in the brain cells of the patient's subthalamic nucleus during the game. The experiments showed that when patients were faced with a decision with 50/50 odds, the pattern of electrical activity in the cells of their subthalamic nucleus reveals their choice about 500 milliseconds before they act on it.

After their surgeries, patients engaged in the same gambling game. This time, Patel et al. specifically targeted the decision-related activity in their subthalamic nucleus with DBS. This caused the patients to make fewer risky decisions in the game. The experiments show DBS can change decision-making behavior in humans. Newer DBS technology may be even more effective at treating brain disorders and cause fewer side effects. Further study into how the brain processes information will also help scientists to better target DBS and possibly treat a broader range of diseases.
DOI: https://doi.org/10.7554/eLife.36460.002

The STN is also an important deep brain stimulation (DBS) target for the treatment of movements disorders such as Parkinson's Disease (PD). DBS surgery provides one of only a few opportunities to record neuronal responses in humans subjects engaged in cognitive tasks (*Zénon et al., 2016*; *Herz et al., 2016*; *Cavanagh et al., 2011*; *Zaghloul et al., 2012*). In addition, researchers can leverage implanted DBS electrodes as a neuromodulation tool to study the role of the STN in an extra-operative setting. As such, numerous researchers have utilized this approach to interrogate the function of the STN in conflict (*Frank et al., 2007*; *Schroeder et al., 2002*), decision-making (*Seymour et al., 2016*; *Seinstra et al., 2016*; *Wylie et al., 2010*; *Zaehle et al., 2017*; *Cavanagh et al., 2011*), and emotional processing (*Le Jeune et al., 2008*; *Le Jeune et al., 2009*).

Studies of STN stimulation using current generation DBS systems have been limited by the systems' design to deliver continuous stimulation. This is a critical limitation to using DBS to explore dynamic aspects of cognition and might be an important source of variability on DBS effects reported in the literature. We hypothesized that targeting stimulation to specific temporal windows during the evolution of a cognitive process (e.g., decision-making), rather than delivering long periods of continuous stimulation, will be critical to understanding the cognitive function and developing new neuromodulation therapies going forward.

In this study, we explore the role of the STN in making decisions under conditions of uncertainty. We employed single-neuronal recordings and intermittent electrical stimulation in human subjects while they engaged in a financial decision-making task (*Patel et al., 2012*). We find that the STN is selectively activated during a brief window for high-uncertainty trials from single-neuronal data. To assess the role of the STN in decision-making during this brief temporal window, we built a custom device which allowed us to precisely deliver intermittent stimulation within short temporal windows. We found that a brief, high-frequency stimulation pulse delivered prior to the choice period promoted a reduction in risk-seeking behavior. To the best of our knowledge, this is the first study to

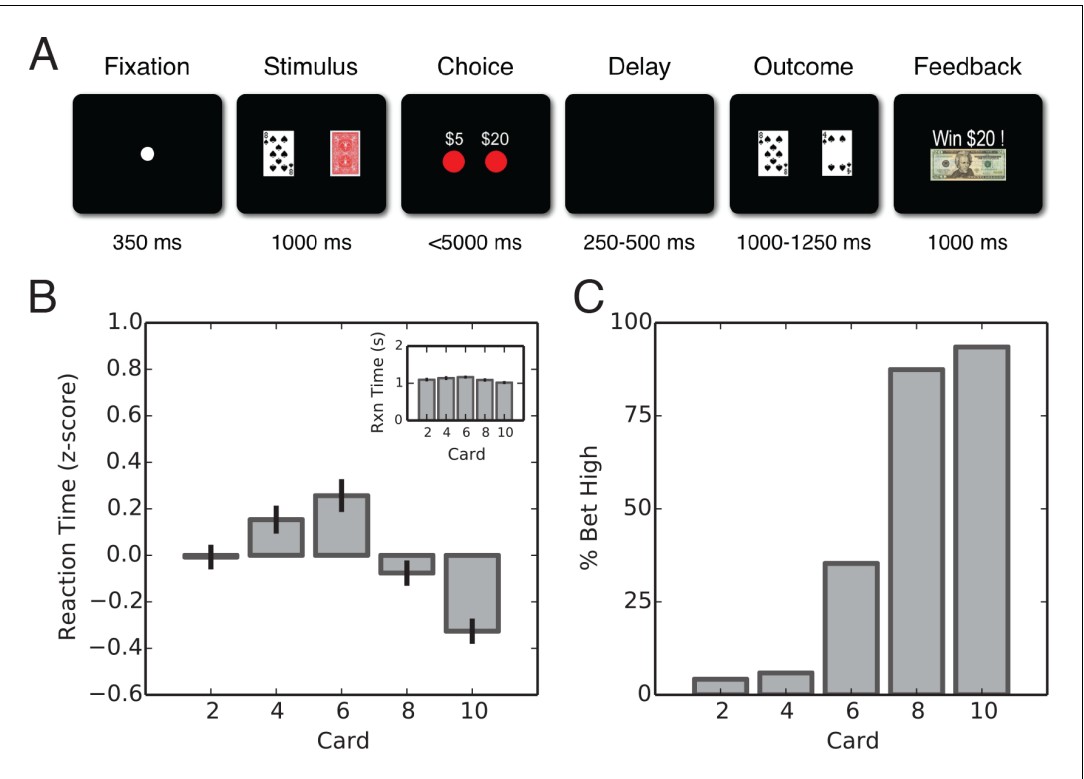

**Figure 1.** Task description and behavioral results. (**A**) Schematic representation of the gambling task. A fixation point is presented to indicate the start of the trial. Next, the subject's card is presented alongside with the back of the computer'scard. Subjects are then presented with the option of placing a $5 or $20 wager. Following a delay, the computer's card is revealed and feedback is presented. (**B**) Average z-scored and raw (inset) reaction times by card for intraoperative population ($F_{4,10} = 10.2$, $p = 4.0 \times 10^{-4}$; ANOVA). Reaction times were the longest for the high-uncertainty trials and amongst the lowest for the low-uncertainty trials. (**C**) Average percentage of high wagers by card value. Intraoperative subjects displayed a strong risk-averse bias that was particularly noticeable on high-uncertainty trials in which a high wager was placed only 24% of the time on average, deviating significantly from a 50/50 strategy ($\chi^2_{1,11} = 42.24$, $p = 1.44 \times 10^{-5}$).

DOI: https://doi.org/10.7554/eLife.36460.003

The following source data is available for figure 1:

**Source data 1.** SQLite database containing two tables: behavior and spikes.

DOI: https://doi.org/10.7554/eLife.36460.004

apply intermittent DBS in humans actively engaged in a cognitive task and the first to demonstrate a reduction in risk-seeking behavior following STN DBS.

## Results

The task is analogous to the classic card game, *War*. Each player was dealt a card – the player with the highest card won (*Figure 1a*). Subjects were first presented with their card. They were then prompted to make either a $5 or $20 wager based on the perceived value of their hand. After a wager was selected, the opponent's hand was revealed followed by visual feedback on the outcome of the trial. For each trial, subjects either won or lost the wagered amount. To simplify the game, we reduced the deck to even cards from 2 through 10 of one suit. Thus, if a subject was dealt a 10-card, the optimal choice would be to place a $20 wager as the outcome is likely to be positive or at worst a draw. Conversely, if the subject received a 2-card the optimal choice would be to place a $5 wager. Uniquely, there is no optimal strategy for the 6-card – the outcome is probabilistically equal.

**Table 1.** Summary table for neuroimaging and intraoperative study populations.
Mean and standard deviation data are given for 24 healthy control subjects (19 male, 5 female) and 6 Parkinson's Disease patients (five men, one female).

| | | Mean | Standard Deviation |
|---|---|---|---|
| Neuroimaging (n = 24) | Age (years) | 36 | 7.5 |
| Intraoperative (n = 6) | Age (years) | 63.2 | 6.8 |
| | Disease Duration (years) | 8.2 | 3.3 |
| | Levodopa dose (mg, daily) | 530 | 300 |

DOI: https://doi.org/10.7554/eLife.36460.005

## Signals of decision-making in the STN

We collected behavioral and neurophysiological data from six subjects (five men, one woman; $63.2 \pm 6.8$ years old; mean $\pm$ Sc.D.; *Table 1*) that underwent DBS surgery for PD. On average subjects performed 1.83 sessions of the gambling task with an average of 105.2 trials per session.

The gambling task was designed such that on any given trial a positive outcome was probabilistically weighted by the subject's card. As such, we expected an engaged participant to display longer reaction times for trials in which the outcome was unpredictable; whereas on predictable trials we expected behavior to converge to an optimal strategy resulting in shorter reaction times. We found such a trend ($F_{4,10} = 10.2$, $p = 4.0 \times 10^{-4}$; ANOVA; *Figure 1b*). Specifically, 6-card trials had the highest average reaction time ($1.16 \pm .19$s and $1.33 \pm .61$s; mean $\pm$ Sc.D., respectively) consistent with the unpredictable nature of the outcome (i.e. an equal chance of winning and losing). Similarly, reaction times for the most predictabletrials were amongst the lowest.

First, we examined behavior on trials in which subjects were dealt a 6-card. A behavioral deviation from a 50/50 betting strategy on these trials would indicate a risk-seeking or risk-averse bias. Overall, we found that subjects had a risk-averse bias placing a high wager only 24% of the time ($\chi^2_{1,11} = 42.24$, $p = 1.44 \times 10^{-5}$; *Figure 1c*). In this study, all intraoperative subjects were off dopaminergic medications at least 12 hr prior to surgery. The low-dopamine state may have contributed to subject's risk-avoidant behavior (*St Onge et al., 2011*; *Claassen et al., 2011*).

Current models suggest that STN activity inhibits responses during cognitively demanding situations (*Frank, 2006*; *Frank et al., 2007*). This inhibition may serve to allow for additional time to process internal and environmental information before ultimately arriving at and executing a decision. To explore this hypothesis in our study we leveraged the intrinsic symmetry of the behavioral paradigm, and divided trials into low and high cognitive demand. The 10- and 2-cards are extreme situations in which the player is probabilistically likely or unlikely to win, respectively — we call these low-uncertainty trials. Conversely, on the 6-card trials the player has an equal probability of winning and losing and there is no optimal strategy — we call these high-uncertainty trials.

We examined single-neuronal data from the STN using standard stereotactic and intraoperative microelectrode mapping procedures. We collected 27 well-isolated neurons with an average of $3.1 \pm 1.1$ (mean $\pm$ Sc.D.) neurons per subject (*Figure 2—figure supplement 1*). All analyses were performed on normalized and pooled spiking data. We *apriori* selected a 500 ms window during the choice period based on previous findings (*Patel et al., 2012*) and explored the relationship between STN activity and the level of uncertainty on a given trial. To do this, we applied a regression model predicting z-scored spike counts as a function of the card value and wager. Interestingly, we found an interaction effect between card value and wager ($F_{9,1450} = 2.55$, $p = 0.02$; ANOVA) but no main effects for card value ($F_{9,1450} = 1.69$, $p = 0.14$) or wager ($F_{9,1450} = 2.68$, $p = 0.10$). Further exploration revealed a significant effect on high-uncertainty trials ($t_{1450} = -2.38$, $p = 0.01$; xtbfFig. 2a; *Figure 2—figure supplements 2, 3* and *4*) which was not present on low-uncertainty trials ($t_{1450} = -1.02$, $p = 0.30$; $t_{1450} = -0.16$, $p = 0.86$; 2- and 10-cards respectively; *Figure 2b*). Interestingly, we found trending activity for the 4- and 8-trials ($t_{1450} = -1.730$, $p = 0.08$; $t_{1450} = -1.78$, $p = 0.07$) which contain an intermediate degree of uncertainty. No other stimulus epoch correlated with subject behavior (*Table 2*).

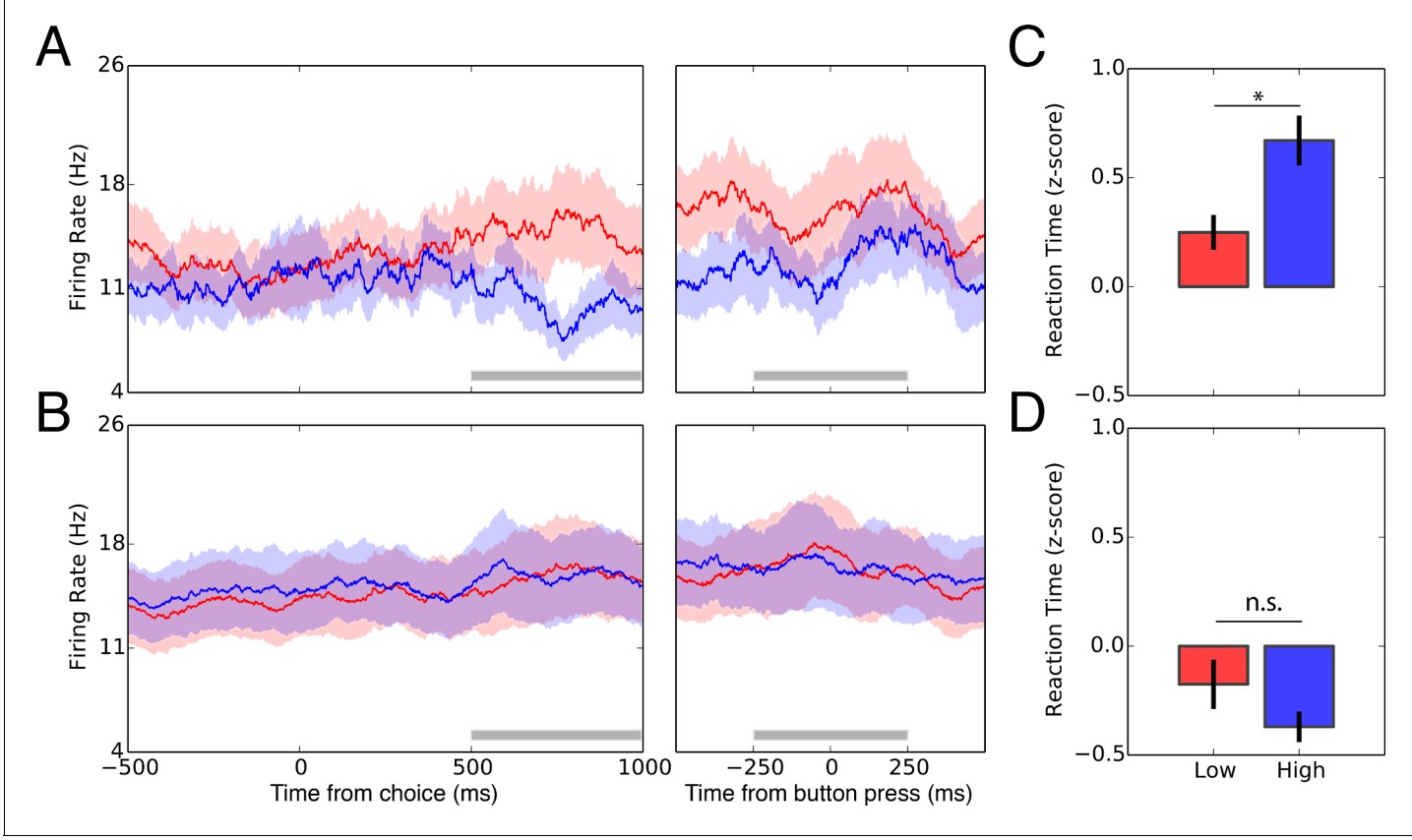

**Figure 2.** Single-neuron activity of decision signaling in the STN. (A) Peri-stimulus time histogram for low wagers (red) vs. high-uncertainty trials zeroed to the onset of the choice period (left panel) and button press (right panel). STN activity predicted the upcoming decision during a 500 ms window (gray bar) during the choice period ($t_{1450} = -2.38$, $p = 0.01$) but not during the button press ($t_{1450} = -1.16$, $p = 0.24$). (B) Peri-stimulus time histogram for low wagers (red) and high wagers (blue) on low-uncertainty trials referenced to the onset of the choice period (left panel) and button press (right panel). STN activity did not encode the upcoming decision for low-uncertainty trials during the choice period ($t_{1450} = -1.02$, $p = 0.30$; $t_{1450} = -0.16$, $p = 0.86$) or the button press ($t_{1450} = -1.48$, $p = 0.13$; $t_{1450} = -0.44$, $p = 0.65$). Shaded regions in (A) and (B) represent standard errors. (C) Average z-scored reaction times for low (red) and high (blue) wagers on high-uncertainty trials. Reaction times were longer for high wagers compared with low wagers ($t_6 = -3.28$, $p = 0.01$). (D) Similarly, reaction times were not significantly modulated by the wager on low-uncertainty trials ($t_9 = 1.17$, $p = 0.27$).

DOI: https://doi.org/10.7554/eLife.36460.006

The following figure supplements are available for figure 2:

**Figure supplement 1.** Examples of two well isolated putative STN neurons.

DOI: https://doi.org/10.7554/eLife.36460.007

**Figure supplement 2.** Example raster of low- (red) and high-wagers (blue) on high-uncertainty (6-card) trials centered on the choice period.

DOI: https://doi.org/10.7554/eLife.36460.008

**Figure supplement 3.** Uncertainty signaling in the STN.

DOI: https://doi.org/10.7554/eLife.36460.009

**Figure supplement 4.** STN activity during movement.

DOI: https://doi.org/10.7554/eLife.36460.010

This signal is unlikely to represent an overt finger movement because our task design balances the presentation of the $five and $20 wagers equally to the left- and right-hand side of the screen. Also, we found no difference in activity between wagers centered on the button press ($F_{9,1450} = 0.24$, $p = 0.98$; *Figure 2a,b*) suggesting this signal was not movement-related. In addition, there was no relationship between the wager ($t_{23} = 0.09$, $p = 0.92$) or the outcome ($t_{23} = 0.71$, $p = 0.48$) on the previous trial.

**Table 2.** Summary of neural task responses.
t-tests were performed for differences in neural responses across task epochs between low- and high-uncertainty trials. Windows of comparison, mean differences, t-values, and p-values are reported for the population with 13 degrees of freedom.

| Epoch | Window | Δ | T | P |
|---|---|---|---|---|
| Fixation | 0–500 ms | −1.52 | 1.25 | 0.23 |
| Card | 0–500 ms | −1.34 | 0.70 | 0.49 |
| Choice | 0–500 ms | −1.79 | 1.31 | 0.21 |
| Choice | 500–1000 ms | −3.75 | 2.96 | 0.01 |
| Feedback | 0–500 ms | 0.17 | −0.17 | 0.86 |

DOI: https://doi.org/10.7554/eLife.36460.011

Lastly, we found that z-scored reaction times on high-uncertainty trials were longer when subjects placed a high vs. low wager ($t_6 = -3.28$, $p = 0.01$; *Figure 2c*). There was no difference in reaction times on low-uncertainty trials for high vs. low wager ($t_9 = 1.17$, $p = 0.27$; *Figure 2d*).

## Effects of intermittent STN stimulation on behavior

We have shown that STN activity within a brief temporal window during the choice period predicts the upcoming wager selectively for high but not low-uncertainty trials. Interestingly, previous human neurophysiology studies have described similar conflict signals arising earlier during the stimulus presentation epoch (*Zaghloul et al., 2012*; *Sheth et al., 2012*). To explore this discrepancy, we used intermittent DBS to test whether altering STN activity during this finite time window would alter the subject's ultimate decision using intermittent DBS. We recruited 13 subjects (12 men, one woman; $62.6 \pm 7.4$ years old; mean ± Sc.D.; *Table 3*) who had previously undergone STN DBS surgery for PD. All subjects had completed surgery at least 6 months prior tithe study.

Through patients' implanted DBS electrodes we applied intermittent electrical stimulation to the STN while subjects were engaged in the same gambling task. Specifically, we applied one of three different stimulation categories randomly on 6-card trials, either giving: no stimulation, 1 s of stimulation during the fixation epoch, or 1 s of stimulation prior to the choice period. To control for observational effects of turning on/off the stimulator (e.g. feeling a sensation when the stimulator turns on), we systematically lowered the voltage setting—blinded to the subject—to a sub-threshold level prior to each experimental session. In addition, we characterized the latency from the trigger to current delivery and found it to be $174 \pm 0.002$ ms ($n=26$; mean ± Sc.D.; *Figure 3—figure supplement 1*). All other settings (e.g. electrode contacts, frequency, and pulse-width) were unaltered from therapeutic levels and were returned to normal following the study.

**Table 3.** Summary table for intermittent stimulation study population.
Mean and standard deviation data are given for 13 subjects (12 men and one woman) who participated in the intermittent stimulation study.

| | Mean (n = 13) | Standard Deviation |
|---|---|---|
| Age (years) | 62.6 | 7.4 |
| Disease Duration (years) | 15.5 | 5.6 |
| Time since surgery (years) | 3.9 | 2.5 |
| Levodopa dose (mg, daily) | 575 | 310 |
| Therapeutic left voltage (volts) | 2.9 | 0.8 |
| Therapeutic right voltage (volts) | 2.9 | 0.7 |
| Therapeutic frequency (Hz) | 180 | 14.3 |
| Study voltage (volts) | 1.0 | 0.9 |

DOI: https://doi.org/10.7554/eLife.36460.012

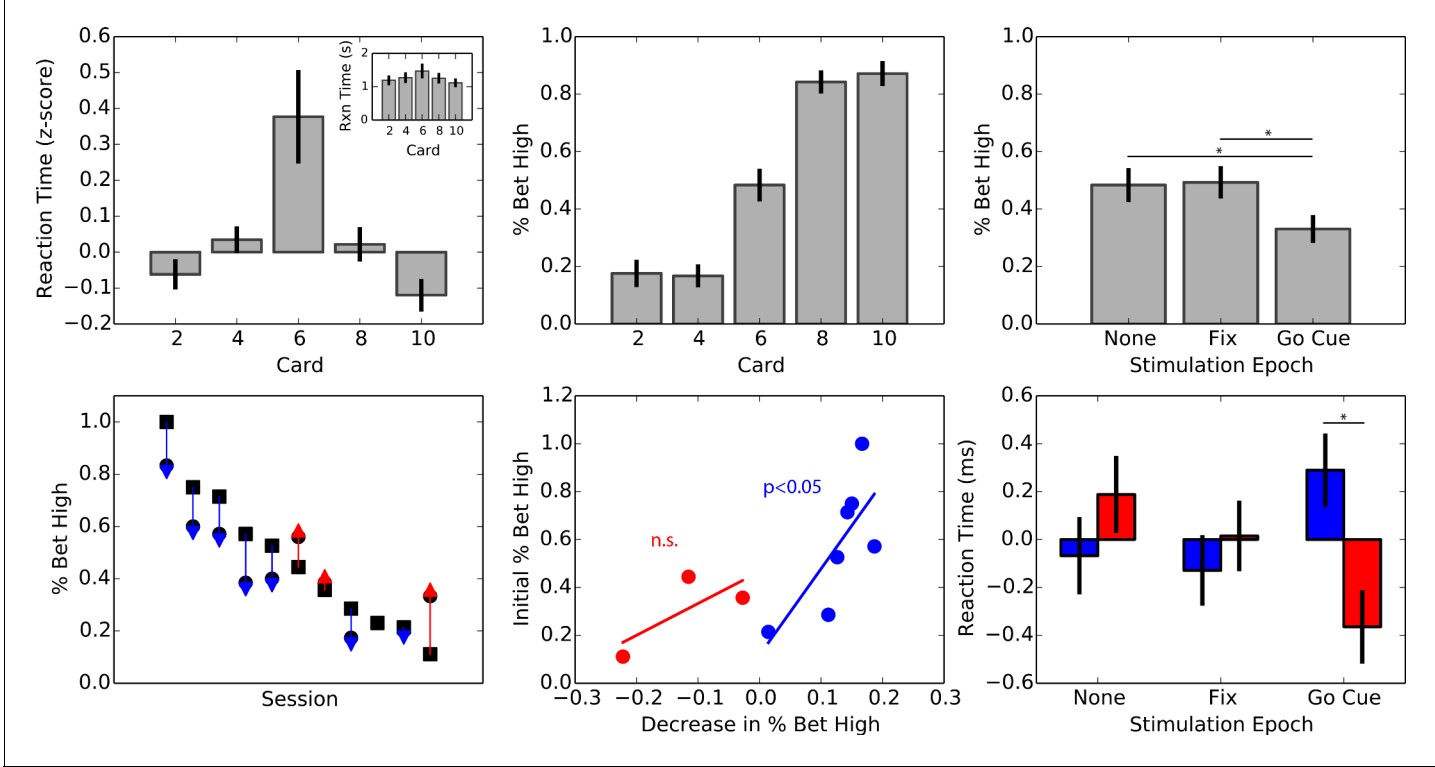

**Figure 3.** Effect of intermittent stimulation on decision-making in the STN. (A) Average z-scored reaction times by card ($F_{4,26} = 5.83$, $p = 0.0002$; ANOVA) and average raw reaction times (inset). Reaction times were the longest for the high-uncertainty trials and the lowest for the low-uncertainty trials. (B) Average percentage of high wagers by card value. Subjects did not significantly deviate from a 50/50 strategy ($\chi^2_{1,27} = 34.13$, $p = 0.13$). (C) Bet high percentage by intermittent stimulation condition. Subjects displayed risk-averse behavior when stimulation was delivered prior to the choice period, placing a high wager 15% less than when stimulation was omitted ($\chi^2_{1,11} = 42.24$, $p = 1.44 \times 10^{-5}$). (D) Bet high percentage on the no stimulation condition (square marker) and on the choice period condition (circle). The arrow indicates the direction of change, risk-averse (blue) or risk-seeking (red). 8/11 subjects displayed a trend towards risk-averse behavior. (E) Scatter plot of percentage change in high wagers during the choice period against high wager percentage on the no stimulation condition, for increases (red) and decreases (blue) in high wagers. Subjects that tended to place a high wager in the baseline (no stimulation) condition tended to experience the greatest change with stimulation. (F) Average z-scored reaction times for low (blue) and high (red) wagers by stimulation condition. No overall effect of stimulation was observed on reaction times ($F_{2,577} = 1.37$, $p = 0.25$; ANOVA) or wager ($F_{1,577} = 0.31$, $p = 0.57$; ANOVA). However, there was a difference in reaction times for high vs. low wagers selectively during the choice period ($t_{22} = 3.72$, $p = 0.001$).

DOI: https://doi.org/10.7554/eLife.36460.013

The following source data and figure supplement are available for figure 3:

**Source data 1.** SQLite database containing a single table: behavior.
DOI: https://doi.org/10.7554/eLife.36460.015

**Figure supplement 1.** Stimulator latency profile.
DOI: https://doi.org/10.7554/eLife.36460.014

On average subjects performed 2 sessions of the gambling task with an average of 108 trials per session. Similar to the intraoperative experiment, we found that subjects demonstrated understanding of the underlying structure of the task ($F_{4,26} = 5.83$, $p = 0.0002$; ANOVA; *Figure 3a*). The fastest reaction times were observed for low-uncertainty trials ($1.19 \pm .76$ seconds, $1.11 \pm .71$ seconds; mean $\pm$ Sc.D.; 2- and 10-cards respectively); and on average, the high-uncertainty trials wreathe slowest ($1.46 \pm 1.16$ seconds; mean $\pm$ Sc.D.). Unlike during the intraoperative sessions, subjects were on their clinical regimen of dopamine replacement therapy during this experiment. We did not observe the same risk-averse behavior on 6-card trials ($\chi^2_{1,27} = 34.13$, $p = 0.13$; *Figure 3b*).

Guided by our neurophysiological findings, we expected that modulation of intrinsic decision signaling prior to the choice period would selectively bias subject behavior. As such, we expected no difference when stimulation was delivered during the fixation period compared to when it was

**Table 4.** Stimulation effect on individual subjects.
Fraction of high-wagers when stimulation was omitted and applied prior to the choice period. The change in decision-making is highlighted blue for a decrease and red for an increase in risk-seeking choices.

| No stim | pre-Choice | Change |
|---|---|---|
| 1.00 | 0.83 | blue-0.16 |
| 0.75 | 0.60 | blue-0.15 |
| 0.71 | 0.57 | blue-0.14 |
| 0.57 | 0.38 | blue-0.18 |
| 0.52 | 0.40 | blue-0.12 |
| 0.44 | 0.56 | red0.11 |
| 0.35 | 0.38 | red0.02 |
| 0.28 | 0.17 | blue-0.11 |
| 0.23 | 0.23 | 0.00 |
| 0.21 | 0.20 | blue-0.01 |
| 0.11 | 0.33 | red0.22 |

DOI: https://doi.org/10.7554/eLife.36460.016

omitted. The data confirmed this hypothesis ($F_{2,28} = 2.93$, $p = 0.05$; ANOVA). In contrast, when stimulation was delivered prior to the choice period, we found that on average subjects had a strong risk-averse bias and placed a high wager only $33.0 \pm 4.83\%$ (mean $\pm$ s.e.m.) of the time, on average an absolute 15% less than the no stimulation group ($t_{28} = 2.77$, $p = 0.009$; **Figure 3c**). Importantly, there was no difference between the omitted and fixation stimulation conditions ($t_{28} = 0.14$, $p = 0.88$), on which subjects placed a high wager on average of $48.3 \pm 5.92\%$ and $49.2 \pm 5.6\%$ of the time (mean $\pm$ s.e.m.), respectively.

To further explore the effects of intermittent stimulation on decision-making we more closely examined the effects within individual subjects. To do so, we plotted each subject's average high wager percentage when stimulation was omitted and delivered at the choice period (**Figure 3d**). Overall, we found subjects spanned a large range in baseline tendency for placing high wagers, ranging from 11% to 100%. We found that 7 out of 11 subjects displayed a reduction in risk-seeking behavior (**Table 4**). Of the seven subjects the average magnitude of change was 12.8% (range: 1–18%). Interestingly, we observed that the magnitude of the reduction in risk-seeking behavior correlated with their initial starting point ($t_7 = 2.46$, $p = 0.05$; **Figure 3e**). For the three subjects that showed an increase in risk-seeking behavior, the average magnitude of change was 12.1% (range: 2–22%). The same correlation did not appear to exist in the this group ($t_2 = 1.13$, $p = 0.46$), though the sample size is limited. One subject experienced no change in either direction from the stimulation.

Lastly, we explored whether stimulation had an effect on subject's reaction time performance. We found that there was no overall main effect of stimulation epoch ($F_{2,577} = 1.37$, $p = 0.25$; ANOVA) or wager ($F_{1,577} = 0.31$, $p = 0.57$; ANOVA) on reaction time. However, we did find an interaction effect between stimulation epoch and wager ($F_{2,577} = 4.15$, $p = 0.01$; **Figure 3f**). Specifically, during the choice period stimulation condition, reaction times were faster when subjects placed a high wager compared with a low wager ($t_{22} = 3.72$, $p = 0.001$), supporting previous findings (**Frank et al., 2007**). No similar differences were observed for the omitted and fixation stimulation conditions.

## Discussion

We used a multi-modal approach consisting of single-neuronal recordings and intermittent stimulation to characterize the neurophysiological role of the STN in decision-making under uncertainty. To do so, we used a financial decision-making task designed to interrogate risk-taking behavior. Using this task, we categorized trials into high and low-uncertainty. We defined high-uncertainty as trials in which the probability of a positive and negative outcome are equal. As a result there was no optimal behavioral strategy. Conversely, low-uncertainty trials were cases in which the outcome was heavily

biased towards or against a positive outcome. On these trials, subject behavior was reliably stereo-typed towards the most appropriate wager to maximize gains or minimize losses. We found that on high-uncertainty trials STN neural activity encoded the upcoming decision in a discrete 500 ms temporal window immediately before the choice period. In a recent functional imaging study, Fleming et al. found a bilateral increase in BOLD response localized to the STN selectively for high-uncertainty trials where subjects responded against a status-quo bias. Although their study uses perceptual decisions, we demonstrate that the same underlying mechanism may extend to value-based decisions made under conditions of uncertainty. Other studies have observed similar neural responses in the STN following conflict related encoding (*Cavanagh et al., 2011*; *Zaghloul et al., 2012*) and control signal encoding (*Isoda and Hikosaka, 2008*; *Wiecki and Frank, 2013*).Unfortunately, the task design in the present study does not let us dissect the influence of conflict, control, and uncertainty on the observed neural responses reported here. This remains an open question within the STN literature body. It is worth noting the possibility that the observed STN neural response in this experiment is a combination of conflict and control. More specifically, a departure from a prepotent response (i.e. placing a high wager) induces STN activity and allows for the recruitment of control centers to mediate a new decision. This would be supported by computational models (*Wiecki and Frank, 2013*) and experimental data (*Coulthard et al., 2012*) but require further investigation to tease apart.

Furthermore, Cavanaugh et al. have previously shown that increases in local-field potential oscillations in the medial prefrontal cortex and STN correlate with trial-by-trial decision conflict and that continuous electrical stimulation through implanted DBS electrodes can prevent adjustments in decision thresholds ultimately resulting in rapid or impulsive decision-making (*Cavanagh et al., 2011*). In contrast, in our data the application of intermittent DBS prior to the choice period resulted in an increasein risk-averse decisions and in reaction times for those decision. One potential explanation for these seemingly conflicting findings is that the effects of stimulation may vary depending on the duration of stimulation. It has previously been suggested that short bursts of high-frequency STN stimulation serve to increase local firing rates which are subsequently silenced with prolonged stimulation (*Lee et al., 2009*). Continuous, high-frequency stimulation has also been proposed to act as an informational lesion, essentially overwriting the normal time-varying activity of the target (*Herrington et al., 2016b*). Our finding also appears to correspond to the observed neurophysiological data from this study, where a slight increase in overall STN activity during the choice period correlates with placing a low wager. An interesting limitation of the stimulation study is that stimulation was only delivered on the high-uncertainty trials, limiting our ability to understand the constraints of its effect on modifying behavior on medium- or low-uncertainty trials. We would hypothesize the effect would be limited or not present on low-uncertainty trials given that no differential encoding was observed, however this remains to be studied.

Interestingly, our findings differ from other human neurophysiology studies in which conflict activity was observed during the stimulus presentation, as opposed to the choice period, in the dorsal anterior cingulate (*Sheth et al., 2012*) and the STN (*Zaghloul et al., 2012*). To further explore the temporal dynamics of the observed signal, we performed a second experiment in which we applied intermittent STN stimulation through implanted DBS electrodes selectively during high-uncertainty trials. Stimulation was delivered either during the fixation period, choice period, or it was omitted. This technique is uniquely different than previous studies using DBS as a method to interrogate neural circuits because we implemented a system for rapidly turning on and off the implanted device, permitting us to time-lock delivery to specific task-epochs. This approach may further reduce confounding effects of long-term stimulation, such as carry-over effects. As a result, we found that intermittent stimulation prior to the choice period—the same interval during which we observed the neurophysiological decision signal from the first experiment—selectively altered subject behavior. No differences were observed in subject behavior when stimulation was omitted or delivered during the fixation period. We found that stimulation prior to the choice period interrupted subjects' ability to appropriately slow responses when betting against their bias (i.e. when they placed a high bet), resulting in a shortened reaction time, consistent with previous work (*Frank et al., 2007*; *Cavanagh et al., 2011*).

Although we attempted to reduce confounding effects by demonstrating both physiological and stimulation evidence to support our claim, our experimental design has several fundamental limitations. In the first experiment, we perform intraoperative recordings in patients undergoing a

neurosurgical procedure. Naturally, there are several limitations for performing studies in the operating room, such as the length of each experimental session. For this reason, the total number of trials and neurons we are able to record can often be limited. In this study, we focus our neurophysiological findings to population responses. Despite this limitation, however, we find the reported effects to be consistent across the population. In addition, we compensate for this limitation by developing a novel stimulation method to carefully test the relationship between our neurophysiological findings and subject behavior. Furthermore, the subjects in this study all suffer from advanced PD, a disease known to affect natural reward processing. For obvious reasons, these experiments are constrained to populations requiring neurosurgical treatment, and direct comparisons to a healthy population are limited to behavioral measures.

In conclusion, we provide functional imaging and neurophysiological evidence in human subjects demonstrating the critical role of the STN in encoding decisions under conditions of uncertainty. Moreover, we demonstrate that electrical stimulation of the STN within a finite temporal window can selectively bias subject behavior towards more risk-averse decisions. Together, this provides evidence for the role of precision neuromodulation approaches and closed-loop deep brain stimulation for the advancement of neurological and neuropsychiatric therapies.

## Materials and methods

### Study subjects

We recruited six subjects undergoing STN DBS for the treatment of Parkinson's disease to participate in the intraoperative neurophysiology study. Each individual was evaluated and considered for surgery by a multidisciplinary team of neurologists, neurosurgeons, and psychiatrists. Once approved and scheduled for surgery an independent member of the research team approached each patient to describe the possibility of study inclusion. At that time risks and benefits were clearly addressed to each subject. All study subjects enrolled voluntarily and provided informed consent under guidelines approved by the Massachusetts General Hospital Institutional Review Board. Subjects were free to withdraw from the study at any time without consequence to operative approach or clinical care. This study was approved by the Massachusetts General Hospital Institutional Review Board (protocol number 2001P000877). For a more detailed description on performing cognitive studies with microelectrode recording during DBS, see (*Patel et al., 2013*).

### Task presentation

A computer monitor was fixed to an adjustable arm and mounted to the operating bed and positioned comfortably within the viewing distance of the patient. A button box was similarly mounted to the operating bed and placed comfortably under the patient's right hand. Subjects were in a comfortable reclined position. The behavioral task was presented using custom written software in Matlab (Math works, Natick, MA), Monkey logic (www.monkeylogic.org) (*Asaad and Eskandar, 2008a*; *Asaad and Eskandar, 2008b*; *Asaad et al., 2013*).

The task is analogous to the classic card game, *War*. On each trial, the subject and computer are each dealt a card and the player with the higher card wins. To simplify the game the deck is limited to five cards: even cards from 2 through 10 from one suit. The rules were carefully explained to each subject prior to the study. Each trial requires the subject to evaluate his/her card, determine its value, and place a $five or $20 wager with the goal of maximizing profits. Thus, when the subject is dealt a 10-card, the optimal choice is to place a $20 wager as the outcome is likely positive or at worst a draw. Conversely, the optimal choice for a 2-card is to place a $five wager since the outcome is likely negative or at best a draw. There is no optimal strategy for the 6-card—the outcome is probabilistically equal.

Each trial began with a fixation point presented at the center of the screen for 350 ms to indicate the start of trial (*Figure 1a*). Next, the subject's card and the back of the opponent's card were displayed for 1000 ms. Two red circles then appeared, indicating the mapping of each button (left and right buttons) to its respective wagers ($five and $20). The button map was presented randomly such that the $five and $20 wagers are assigned to the left and right buttons equally. The presentation of the button map also serves as the choice period, indicating when to initiate a wager. The time it took the subject to press a button was considered the reaction time with a maximum of 5 s.

Following the wager, there was a randomized delay period of 250–500 ms, which was immediately followed by the presentation of the subject's and computer's card for 1000–1250 ms. Lastly, feedback was given for 1000 ms by displaying an image of a $five or $20 bill with text indicating the outcome. In the case of a draw, only text is displayed. Subjects were monetarily rewarded following their participation in the study.

## Electrophysiology

For a detailed description, please see (*Patel et al., 2013*). Intraoperative microelectrode recordings were performed using three Para-sagittal tungsten microelectrodes (*Figure 1b*). The electrodes were advanced using a motorized Alpha Omega (Alpha-Omega Engineering, Nazareth, Israel) Microdrive. Intraoperative motor testing was performed at <1 mm increments throughout the dorso-lateral-ventromedial axis of the STN to characterize the motor and non-motor compartments. Recordings were band-pass filtered between 300 Hz and 6.5 kHz by an Alpha Omega acquisition system. Data was recorded at 20 kHz by a PowerLinc 1401 acquisition system (Cambridge Electronic Design, Cambridge, England) and stored for post-hoc analysis. Offline, the neurophysiology data was sorted into individual neuronal records using a template clustering method (Offline Sorter, Plexon, Houston, TX). Data from each electrode was sorted separately.

## Behavioral and neuronal analysis

All analyses were performed using a combination of iPython and R. Because of inter-subject variability in baseline motor performance, we explored reaction time differences by first z-scoring data using each session's mean and variance reaction time. Normalized subject data and allowed for equal comparisons for group level analyses. We then applied either a one-way or two-way ANOVA on the z-scored reaction time data to assess statistical differences. Post-hoc analyses were performed using two-tailed $t$-tests.

To visualize neural activity, the instantaneous firing rate was approximated by convolving a Gaussian kernel (sigma = 150 ms) with 1 ms binned spike trains. Because of the limitations in the number of trials recorded in each experimental session, statistical analyses at the individual cell level were rarely significant, and instead all analyses were performed at the population level. Statistical differences between population responses were assessed using two-tailed $t$-tests during pre-defined 500 ms windows: 500–1000 during the choice period and −250–250 centered on the button press based on a previous study (*Patel et al., 2012*).

To explore the relationship between neural activity and the decision, we applied a linear regression model of the form: $Z = \beta_0 + \beta_c C + \beta_w W + \beta_{cw} C \times W$, where $Z$ is a vector of z-scored spike counts (relative to each neuron) in a 500 ms window, $C$ is the card value, $W$ is the wager, and $C \times W$ the interaction between the two terms. Both $C$ and $W$ are categorical variables and represented with dummy variables in the regression model. Coefficients were estimated through a least-squares approach.

## Intermittent stimulation

Thirteen study participants were recruited from STN DBS patients identified by their movement disorders neurologist to participate in the intermittent stimulation study. A study staff member contacted potential study participants by telephone to introduce the study and invite the patient to participate. On the day of the study, after obtaining written informed consent, the patient's deep brain stimulator was turned off. Subsequently the stimulation voltage was lowered in small increments with the stimulator being turned on and off in a blinded fashion until a voltage threshold was reached at which the patient was unable to detect the stimulation. The stimulator controller was secured over the patient's pulse generator, and after approximately 15 min with the stimulator off, the patient began playing the task.

The task was conducted during the day in a quiet room. Patients were permitted to take short breaks as needed during the task. We used three different stimulation conditions on 6-card trials: 1 s of stimulation at the fixation epoch, 1 s of stimulation at the choice period, or no stimulation was delivered. This design allowed each subject to act as his own control, helping to account for variance due to differing disease, medication, and electrode location factors between patients and also

allowed us to control for general versus time specific effects of stimulation. This study was approved by the Massachusetts General Hospital Institutional Review Board (protocol number 2007P001806).

## Acknowledgments

We are grateful to the patients that participated in this study. This work was supported by grants from the National Institutes of Health (NEI 1R01EY017658-01A1, NIDA 1R01NS063249, NIMH Conte Award MH086400, R25NS065743), the Klingenstein Foundation, Howard Hughes Medical Institute, Dana Foundation, the Sackler Scholar Programme in Psychobiology and the American Brain Foundation/American Academy of Neurology. This research was funded by the Defense Advanced Research Projects Agency (DARPA) under Cooperative Agreement Number W911NF-14-2-0045 issued by the Army Research Office contracting office in support of DARPA'S SUBNETS program. The views, opinions, and/or findings expressed are those of the author(s) and should not be interpreted as representing the official views or policies of the Department of Defense or the U.S. Government.

## Additional information

### Competing interests

Michael J Frank: Senior editor, *eLife*. The other authors declare that no competing interests exist.

### Funding

| Funder | Grant reference number | Author |
| --- | --- | --- |
| Defense Advanced Research Projects Agency | SUBNETS | Shaun R Patel<br>Todd M Herrington<br>Jimmy C Yang<br>Alik S Widge<br>Darin Dougherty<br>Emad N Eskandar |

The funders had no role in study design, data collection and interpretation, or the decision to submit the work for publication.

### Author contributions

Shaun R Patel, Conceptualization, Data curation, Formal analysis, Validation, Investigation, Visualization, Methodology, Writing—original draft, Writing—review and editing; Todd M Herrington, Michael J Frank, Formal analysis, Investigation, Writing—review and editing; Sameer A Sheth, Data curation, Investigation, Writing—review and editing; Matthew Mian, Data curation, Investigation; Sarah K Bick, Jimmy C Yang, Alice W Flaherty, Data curation, Writing—review and editing; Alik S Widge, Writing—review and editing; Darin Dougherty, Data curation, Funding acquisition, Writing—review and editing; Emad N Eskandar, Data curation, Funding acquisition, Investigation, Writing—review and editing

### Author ORCIDs

Shaun R Patel http://orcid.org/0000-0002-9334-2522
Michael J Frank http://orcid.org/0000-0001-8451-0523
Alik S Widge https://orcid.org/0000-0001-8510-341X

### Ethics

Human subjects: Informed consent was obtained from all subjects in this study. This study was approved by the Massachusetts General Hospital Institutional Review Board (protocol number 2001P000877).

### Decision letter and Author response

Decision letter https://doi.org/10.7554/eLife.36460.019
Author response https://doi.org/10.7554/eLife.36460.020

## Additional files

### Supplementary files

• Transparent reporting form
DOI: https://doi.org/10.7554/eLife.36460.017

### Data availability

Data curation and sharing is currently in process for the entire DARPA SUBNETS project. The underlying data for Figures 1, 2, and 3 in this study have been provided.

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
