## [Decision Letter]

Thank you for submitting your article "Intermittent subthalamic nucleus deep brain stimulation induces risk-aversive behavior in human subjects" for consideration by *eLife*. Your article has been reviewed by Richard Ivry as the Senior Editor, a Reviewing Editor, and three reviewers. The following individual involved in review of your submission has agreed to reveal their identity: Birte Forstmann (Reviewer #2).

The reviewers have discussed the reviews with one another and the Reviewing Editor has drafted this decision to help you prepare a revised submission.

This study provides evidence for a role of the subthalamic nucleus (STN) in modulating risk-attitude in humans. This is based on three different experimental methods. First, the authors show that human STN is particularly active during decisions under risk, when uncertainty is high. Second, spiking activity of single STN units are significantly more active for risk-averse choices during high uncertainty, but not low uncertainty trials. Third, short-duration DBS stimulation while subjects make a decision increases the number of risk-averse choices. This last finding indicates that the increased STN activity might be causally related to the shift in risk attitude.

All reviewers thought that the topic of your study addresses a very interesting and timely question, and that your basic experimental design is sound, and sufficient to support the main conclusions. The reviewers liked in particular that you combined a number of different experimental techniques to investigate a common underlying question. That is a great strength of the paper. However, the reviewers also noted a number of weakness that need to be addressed, before the paper can be considered for publication.

Essential revisions:

1) There are considerable reservations among the reviewers regarding the fMRI experiments included in the paper. First, there are issues with the comparability of study populations. All reviewers agreed that you should have tested healthy age-matched controls, since behavioral strategies might change with age. Secondly, there are issues with the MRI scanning protocol and data analyses. The description of the MRI protocol lacks a lot of important information and in light of the recent literature, it appears unlikely that (a) sufficient tSNR, and (b) anatomical specificity to extract signal from the STN is obtained in this study (see, e.g., de Hollander et al., (2017); de Hollander et al., (2015)). This is particular worrisome since voxels were resampled to 2mm voxels, and subsequently smoothed with a three-dimensional Gaussian kernel of 6 mm width (FWHM). In light of these shortcomings, we request that you take out the description of the fMRI experiments from the paper.

2) The exact functional interpretation of the STN activity is not clearly expressed. There are a number of possibilities: (1) a 'conflict' signal that represents the presence of two mutually incompatible response tendencies, (2) 'uncertainty' in the economic sense of risk (i.e., high variability in the response outcome distribution), (3) a 'control signal' that inhibits the risk-seeking response and thus biases choices towards risk-avoiding behavior. Addressing this question requires the inclusion of a computational model that makes clear the author's interpretation of STN activity. Specifically, could choice behavior on the "6" trials be modeled using choice and reward history in such a way as to predict behavior? Such a model might produce parameters that could be more systematically related to observe STN responses and behavior to identify what decision-related parameters might be driving neural activity or modified by stimulation.

3) The basic result that intermittent DBS during the choice period on high-uncertainty trials leads to risk averse behavior is supported by the evidence provided; however, if stimulation had been delivered on all trials (not just 6-card trials), it would have been possible to determine if the effect of stimulation was restricted to conditions with high uncertainty, or if stimulation led to a systematic, consistent bias toward risk averse behavior irrespective of uncertainty. The authors should include such experiments, if possible. If these experiments are too effortful, we strongly suggest that the authors consider this issue as a notable limitation of the study in their discussion, specifically addressing how their interpretation would change depending on what broader patterns might have been observed.

---

## [Author Response]

Essential revisions:1) There are considerable reservations among the reviewers regarding the fMRI experiments included in the paper. First, there are issues with the comparability of study populations. All reviewers agreed that you should have tested healthy age-matched controls, since behavioral strategies might change with age. Secondly, there are issues with the MRI scanning protocol and data analyses. The description of the MRI protocol lacks a lot of important information and in light of the recent literature, it appears unlikely that (a) sufficient tSNR, and (b) anatomical specificity to extract signal from the STN is obtained in this study (see, e.g., de Hollander et al., (2017); de Hollander et al., (2015)). This is particular worrisome since voxels were resampled to 2mm voxels, and subsequently smoothed with a three-dimensional Gaussian kernel of 6 mm width (FWHM). In light of these shortcomings, we request that you take out the description of the fMRI experiments from the paper.

Thank you for raising this important issue. We agree that additional consideration should be taken before interpreting the imaging results. We have removed the description of the functional study from the revised version of the manuscript.

2) The exact functional interpretation of the STN activity is not clearly expressed. There are a number of possibilities: (1) a 'conflict' signal that represents the presence of two mutually incompatible response tendencies, (2) 'uncertainty' in the economic sense of risk (i.e., high variability in the response outcome distribution), (3) a 'control signal' that inhibits the risk-seeking response and thus biases choices towards risk-avoiding behavior. Addressing this question requires the inclusion of a computational model that makes clear the author's interpretation of STN activity. Specifically, could choice behavior on the "6" trials be modeled using choice and reward history in such a way as to predict behavior? Such a model might produce parameters that could be more systematically related to observe STN responses and behavior to identify what decision-related parameters might be driving neural activity or modified by stimulation.

Thank you for raising this important concern. We agree that a computational model dissecting risk vs. uncertainty vs control could aid in providing potential mechanistic underpinnings on the observed neural responses. After considerable thought we do not believe that this task captures the necessary variability required to draw reliable conclusions.

The’ War’ task, as presented, is designed to capture’ uncertainty’ through 6-card trials which represent the experimental condition with the highest decision entropy. A mixed-effects generalized linear model on spiking activity in the 500 ms window during the choice period supports uncertainty encoding with no significant terms on reward history (win/loss on previous trial) or Gratton effect (uncertainty on the previous trial).

Conflict and control encoding are particularly challenging to address with this task design. Conflict requires mutually competing choices which is not explicitly controlled for in this design. Control is similarly difficult as there were no time-to-choice constraints placed on subjects. Other studies have demonstrated both conflict (Cavanaugh et al., 2011, Zaghloul et al., 2012) and control related STN encoding (Isoda and Hikosaka, 2008; Wiecki and Frank, 2013).

We have added this limitation to the Discussion section.

3) The basic result that intermittent DBS during the choice period on high-uncertainty trials leads to risk averse behavior is supported by the evidence provided; however, if stimulation had been delivered on all trials (not just 6-card trials), it would have been possible to determine if the effect of stimulation was restricted to conditions with high uncertainty, or if stimulation led to a systematic, consistent bias toward risk averse behavior irrespective of uncertainty. The authors should include such experiments, if possible. If these experiments are too effortful, we strongly suggest that the authors consider this issue as a notable limitation of the study in their discussion, specifically addressing how their interpretation would change depending on what broader patterns might have been observed.

We completely agree and have previously considered this experiment ourselves. Although we hope to perform this experiment in the future, at the moment, it will need to remain outside of the scope of this manuscript. The lab in which this work was originally performed has recently moved to another university. Additional time and resources will be required to re-initiate these experiments. We have expanded the Discussion to reflect this experimental limitation and elaborate on potential implications.